# The Effect of Sexes and Seasons on the Morphological Structures of the Ruminant Digestive System of Blue Sheep (*Pseudois nayaur*)

**DOI:** 10.3390/ani13061084

**Published:** 2023-03-17

**Authors:** Dehuai Meng, Yuhui Si, Jifei Wang, Zongzhi Li, Romaan Hayat Khattak, Zhensheng Liu, Liwei Teng

**Affiliations:** 1College of Wildlife and Protected Areas, Northeast Forestry University, Harbin 150040, China; 2National Nature Reserve of Nanhua Mountains in Ningxia, Zhongwei 755200, China; 3Key Laboratory of Conservation Biology, National Forestry and Grassland Administration, Harbin 150040, China

**Keywords:** digestive systems, environmental adaptations, Helan Mountains, *Pseudois nayaur*, ruminant

## Abstract

**Simple Summary:**

Ruminants, as widely distributed herbivorous animals, have developed various adaptive nutrition strategies in response to environmental changes. As a typical mountain ruminant, blue sheep have a strong adaptability to the environment. In order to understand the changing patterns in the digestive tracts of blue sheep, the morphological characteristics and physiological indexes of the digestive tract structure were determined. Our conclusion is that blue sheep have adopted the nutritional strategy of roughage feeders. In order to adapt to environmental changes, blue sheep adopt fast feeding and fast excretion modes to allow feeding on large amounts of plants, mainly those with high cellulose content, low nutrient quality, and those that are easy to access. Our results can serve as a basis for studying the digestive system adaptation of other ruminants.

**Abstract:**

Constant adaptation to environmental changes is required by ruminants to allow them to adapt to different ecological niches and feeding habits. In addition, the morphology and function of ruminant digestive systems reveal some adaptive evolutionary characteristics. Blue sheep (*Pseudois nayaur*) display a variety of morpho-physiological adaptations that are typical of grazers. In this study, we collected 64 adult blue sheep samples (whole animal carcasses) from the Helan Mountains, China, during different seasons. The external morphological parameters, digestive system morphological indexes, and rumen surface enlargement factors were determined. Our results reveal that the rumen and reticulum weights were positively correlated with the body weight (*p* = 0.004), while the food channel aperture, intestinal length, and weight of the blue sheep digestive tract presented no significant differences between different seasons (*p* > 0.05) and sexes (*p* > 0.05). There were significant differences (*p* < 0.001) in the density, length, and width of mastoids, and the rumen surface enlargement factor was 2.85 ± 1.37, which is typical of roughage feeders. The nutritional and ecological characteristics of blue sheep represent obvious morphological and physiological adaptations to an herbivorous diet. Adopting a quick foraging strategy allows blue sheep to rapidly consume and excrete large amounts of feed, thus obtaining the required energy for their activities and facilitating better adaption to environmental changes.

## 1. Introduction

Ruminants are widely distributed herbivores with relatively successful evolution in terms of their morphology and digestive systems [1,2,3]. The morphological structure and function of their digestive tract adapts to changes in their feed and the corresponding nutritional adaptation strategy [4]. The ruminant digestive system consists of a long muscular tube that extends from the mouth to the anus and has numerous attached glands. The digestive tube consists of the mouth, pharynx, esophagus, pre-stomachs (reticulum, rumen, and omasum), true stomach (abomasum), small intestine (duodenum, jejunum, and ileum), and large intestine (cecum, colon, and rectum). The stomach includes the rumen, reticulum, omasum, and abomasum, through which the feed particles pass [5]. The first three chambers are known as the pre-stomach, which favors the digestion of the structural carbohydrates that are part of ruminant diets [6,7]. Only the last chamber, the abomasum, is comparable in structure and function to the simple stomach of most other animal species [8].

In the process of evolution, ruminants have interacted with the environment for a long period of time [9]. In order to adapt to different environments and intake sufficient energy, their digestive organs have adapted to different feed resources in terms of their structure and function. For species with large rumens, the feed they ingest has a large space for fermentation, where it remains for a long time, enabling cellulose and other substances in the feed to be fully fermented and decomposed, thereby obtaining nutrients needed for their physiological activities. Species with smaller rumens obtain nutrients by speeding up the flow of feed through the digestive tract and choosing feeds that are more nutritious and easily digestible, which is a physiological phenomenon referred to as the Jarman–Bell principle [10]. The rumen is where most digestion happens and determines the capacity of feed intake among ruminants. The mass of the rumen–reticulum organs, the organs that comprise the largest part of the gastrointestinal tract of ruminants, can vary based on the bulk diet and nutrient availability [11,12,13].

The organs that compose the gastrointestinal tract have high energy demands. Therefore, variations in the weight of these organs have the potential to impact metabolic requirements. According to the characteristics of the digestive tract and feeding habits, herbivores are divided into three types of nutritional adaptation strategies: concentrate selectors, mixed feeders, and roughage feeders [4]. It is believed that the rumen and reticulum of roughage feeders are larger; the proportion of rumen and reticulum contents to the animal’s body weight ranges between 11% and 20%, while that of concentrate selectors is less than 11% [14]. Moreover, the rumen surface enlargement factor (SEF) of roughage feeders is generally less than 3.5, while the SEF of concentrate selectors varies between 7.0 and 9.5, and the SEF of mixed feeders is between the aforementioned ranges [14]. In terms of the digestive system morphology and structure, roughage feeders have a longer intestinal system, while concentrate selectors have a larger large intestine due to the fermentation of cecum [14].

The morphology and function of ruminant digestive systems transform under the impact of seasonal changes, with the aim of adapting to changes in feed resources [15]. The various ruminants adopt different nutritional strategies due to the different proportions of digestible substances in the plants on which they feed. Consequently, the size of the feed particles differs among the stomachs of various ruminants and is one of the most important factors affecting the speed at which feed passes through the digestive tract [16,17], which can directly determine the ratio of nutrients taken in by ruminants and thus affect the digestibility of the feed [18]. Moreover, the velocity of feed particles is related to their size, the type of feed, and the nutritional strategy of the animal [14,19].

The blue sheep is a medium-sized wild sheep [20] (adult blue sheep head to body length = 115–160 cm, tail length = 10–15 cm, shoulder height = 69–91 cm, and weight = 35–75 kg) [21]; this species is rarely active in forests and prefers habitats close to bare rocks and cliffs [22,23,24,25]. The blue sheep is an endemic species of the Qinghai–Tibet Plateau and its adjacent areas, and is widely distributed in China, India, and Pakistan, among other places [22,23,26,27,28,29,30]. Blue sheep are also present in Sichuan, Tibet, Qinghai, Ningxia, and Inner Mongolia, China [22]. The blue sheep is a second-class key protected animal in China and was listed as being of least concern by the International Union for Conservation of Nature (IUCN) in 2000 [31].

The Helan Mountains are the easternmost distribution range of blue sheep in China, featuring the most stable populations [32,33]. Blue sheep is the dominant ungulate species in the Helan Mountains [34,35], and their population density is one of the highest in the world, reaching 3.627~4.635 sheep/km^2^ [36]. This species is one of the main prey animals of the endangered snow leopard (*Panthera uncia*) [37], playing an important role in protecting biodiversity and maintaining the ecosystem balance [38].

By studying the morphological and structural characteristics of the digestive organs of blue sheep and the distribution of feed particles in their digestive tracts, the mechanisms of this species in adapting to different seasons can be evaluated and explored, which will be of great significance to understand their population dynamics and survival strategies. The following conclusions can be drawn from the current study: (1) the morphological and structural characteristics of the blue sheep digestive tract are consistent with the nutritional adaptation strategy of roughage feeders; (2) the digestive system of blue sheep exhibits strong adaptability to environmental changes, and the different grain sizes of feed particles in the digestive tract show changes according to the season. We believe that the results obtained in the current study will provide benchmark information on blue sheep nutritional adaptation strategies.

## 2. Materials and Methods

### 2.1. Study Area

The current study was conducted in the Helan Mountain National Nature Reserve (Hunnert) in Northwest China (E 105°40′–106°41′, N 38°19′–39°22′). The study area is located in a transitional zone between the steppe and desert regions of Central Asia [39], situated between the Alca Plateau and the Yinchuan Plain. The mountains are oriented in a north–south direction, stretching 250 km from north to south, with an average altitude of 2000–3000 m, and the highest peak is 3556.1 m a. s. l. The mountains cover an area of 2740 km^2^ (including the Ningxia Helan Mountain National Nature Reserve (2063 km^2^) and Inner Mongolia Helan Mountain National Nature Reserve (677 km^2^)). The Helan Mountains are adjacent to the Yellow River Loop and the Ordos Plateau; they are adjacent to the Tengger Desert in the west, the Mu Us Sandy Land in the east, and the Ulan Buhe Desert in the north [40]. Snow cover is limited in the Helan Mountains [41] and vegetation distribution is strongly influenced by moisture conditions, exhibiting a typical temperate mountain forest ecosystem. Vegetation is distributed vertically and can be divided into four vertical vegetation zones from the foothills to the main peak [42,43]: (1) mountain steppe belt; (2) mountain sparse forest steppe belt; (3) mountain coniferous forest belt; and (4) subalpine shrub and meadow belt. The annual average temperature is −0.9 °C, and the annual average precipitation is 420 mm [44]. In addition to blue sheep, the Helan Mountains are home to the snow leopard (*Panthera uncia*), alpine musk deer (*Moschus chrysogaster*), red deer (*Cervus elaphus*), and lynx (*Lynx lynx*). This area is one of the key biodiversity zones in China [45]. The map was obtained from the National Catalogue Service for Geographic Information (https://www.webmap.cn/, accessed on 3 September 2022) (Figure 1).

### 2.2. Measurement of External Morphological Characteristics of Blue Sheep

From 2012 to 2013, 64 adult blue sheep samples were collected in the Helan Mountains; the samples were obtained from legal hunting and mainly used for specific research activities such as creating animal specimens and disease investigations. There were 23 individuals for which the ages were known [46] and 47 individuals of known sex (15 females and 32 males). The samples and measurements data were stored in the College of Wildlife and Nature Reserve, Northeast Forestry University (Access number: YY2012_01-64). The measurement indexes were affected by the preservation integrity and collection time of the blue sheep samples.

The body weight (to 0.1 kg precision) and body length, tail length, and ear length (to 0.1 cm precision) were measured. The compound stomach (including stomach contents) was separated from the cardia and pylorus and weighed [47]. We then separated the reticulum, omental stomach, and abomasum from the opening of the reticular valve and the opening of the valve fold, measured the weight (accurate to 0.1 g) of each part of the stomach (including the contents), and measured the diameter of the opening of the reticular valve and the valve fold (to 0.01 cm precision). Finally, the contents of the stomach were poured into containers for use, and the volume of the rumen, reticulum, omasum, and abomasum were measured using the irrigation method, and these tissues were weighed [48]. To measure the intestinal tract of blue sheep, the mesentery was cut down and stretched, and the intestinal tissue was straightened; then, we measured the length of the small intestine, cecum, large colon, small colon, and rectum (to 0.1 cm precision), as well as the lengths of each intestinal tissue and rectum. Finally, the weights of the contents were measured (to the nearest 0.01 g) [49].

### 2.3. Measurement of Rumen Surface Enlargement Factor

Samples were taken from the dorsal and ventral rumen, the atrium ruminis, and the bottom of the dorsal blind sac and preserved in alcohol. Then, the surface enlargement factor (SEF) of the papillae was determined by measuring the number of papillae and the mean height and width per square centimeter [50]. In the upper wall of the rumen, we cut tissue blocks of 5 × 5 cm^2^ from the abdominal wall, the bottom of the vestibule, and the posterior blind sac and stored them in 60% alcohol solution for later use. We then removed a 1 × 2 cm^2^ block of the dehydrated and fixed tissue sample. The number of ruminal mucosal papillae was determined via the counting method under a dissecting microscope. The height and middle width (to 0.01 cm precision) of 20 mastoids were randomly measured, and the SEF value was calculated [51].

### 2.4. Distribution of Feed Particles in Different Parts of Digestive Tract of Blue Sheep

The distribution of feed particles was measured in different parts of the digestive tract (rumen, reticulum, omasum, abomasum, small intestine, cecum, and rectum). The contents of each part were poured into different containers and thoroughly stirred. A mixture of 150 mm was randomly extracted and poured into a sieve drawer (sieve diameter x > 2.0 mm, 1.0 mm < x ≤ 2.0 mm, 0.425 mm < x ≤ 1.0 mm, 0.25 mm < x ≤ 0.425 mm, and 0.125 mm < x ≤ 0.25 mm). The samples in each sieve drawer were dried and weighed (to 0.0001 g precision). The proportion of dry matter weight was calculated. The seasonal differences in the distribution and proportion of feed particles in different parts were compared to verify the nutritional strategy of blue sheep.

### 2.5. Statistical Analyses

Pearson correlation analysis was used to test the external morphological characteristics of the blue sheep, and linear regression analysis was used to test the effect of body length and age on the length of the digestive tract, as well as the effect of body weight on the weight of each gastric tissue sample, the weight of the stomach contents, and the pore size of each feed passage in the reconstituted stomach. The data for the digestive tract morphology and contents were subjected to K-S testing on a single sample. One-way analysis of variance was used to test the effect of sex on the morphological characteristics of blue sheep, the digestive tract length, the weight of each gastric tissue in the reconstituted stomach, the weight of the stomach contents, the aperture of the feed passage, and the characteristics of different regions of the rumen wall. In the model, sex was a factor to be tested. The analysis method used for feed particle distribution and seasonal differences was the same as that outlined above. We further tested the above indicators considering the effect of body length or body weight together with the body weight or body length, respectively, as a covariate using one-way analysis of covariance to control for body length or body weight. 

For samples with known age, one-way variance (ANOVA) and covariance (ANCOVA) analysis were used to test the above indicators, and post hoc tests were used to analyze the characteristic differences of the dorsal wall, abdominal wall, dorsal cecum, and vestibule. SPSS 20.0 (International Business Machines Corporation in Armonk, NY, USA) was used for all statistical analyses. ArcGIS 10.7 (Environmental Systems Research Institute, Inc. in Redlands, CA, USA) was used to produce the study area map. Origin 2021 (OriginLab Corporation in Northampton, Northampton, MA, USA) was used to create the analysis charts.

## 3. Results

### 3.1. External Morphological Characteristics of Blue Sheep

The external morphological parameters of 41 blue sheep (30 males and 11 females) were measured. The one-way variance (ANOVA) and covariance (ANCOVA) analysis showed that there were no significant differences in the body length, or ear length, tail length, and rear foot length between males and females, while other characteristics showed obvious dimorphism (Table 1). Considering the influence of age on some external morphological characteristics, two-way variance and covariance analyses were further used to test the samples. The results showed that males had relatively larger body weights (*F*_1,16_ = 4.770, *p* = 0.044), larger neck circumferences, (*F*_1,16_ = 6.591, *p* = 0.021), and larger horn base circumferences (*F*_1,16_ = 80.787, *p* < 0.001) than females. However, there was no significant difference in the bust size between them (*F*_1,16_ = 0.634, *p* = 0.438) (Table 1).

Pearson correlation analysis on blue sheep with a known age (*n* = 23) revealed that with increasing age, their body weight (*r* = 0.641, *p* < 0.001), horn length (*r* = 0.548, *p* = 0.001), bust (*r* = 0.640, *p* < 0.001), neck circumference, (*r* = 0.497, *p* = 0.003), and horn base circumference (*r* = 0.524, *p* = 0.002) increased significantly, but the remaining characteristics did not change (*p* > 0.05). Body length had a minor relationship with ear length (*r* = 0.203, *p* = 0.204), tail length (*r* = 0.098, *p* = 0.541), and shoulder height (*r* = 0.259, *p* = 0.102), while horn length, hind foot length, bust, neck circumference, and horn base circumference increased significantly with an increase in body length (*p* < 0.032) (Figure 2).

### 3.2. Morphology and Weight Change of the Stomach of Blue Sheep

The weight of the stomach tissue was 0.78 ± 0.03 kg (*n* = 61). The weight of the stomach (including contents) was 3.77 ± 0.28 kg (*n* = 63, Table 2). The tissue weight of the stomach was positively correlated with body weight (*r* = 0.544, *F*_1,23_ = 9.678, *p* = 0.005), but the proportion of stomach to body weight decreased significantly with an increase in body weight (*r* = −0.705, *F*_1,23_ = 22.747, *p* < 0.001, Figure 3a). The stomach tissue weight and stomach weight (including contents) of blue sheep were significantly positively correlated with age (*r* = 0.709, *F*_1,31_ = 31.385, *p* < 0.001; *r* = 0.511, *F*_1,33_ = 11.685, *p* = 0.002). The proportion of the stomach tissue weight to body weight showed a trend of decreasing with age, but this change was not significant (*r*= −0.289, *F*_1,22_ = 1.817, *p* = 0.193, Figure 3b).

One-way ANOVA and covariance analysis showed no significant difference in the stomach tissue weight (ANOVA, *F*_1,42_ = 0.023, *p* = 0.881), stomach weight (including contents, ANOVA, *F*_1,44_ = 0.299, *p* = 0.587), and the proportion of stomach tissue weight to body weight (ANCOVA, *F*_1,24_ = 2.157, *p* = 0.155) between males and females. Multivariate variance and covariance analyses were conducted taking into account the influence of age on the stomach weight (including contents) and the proportion to body weight. The results showed no significant difference in the stomach weight (including contents, *F*_1,17_ = 0.804, *p* = 0.382) and its proportion to body weight (*F*_1,11_ = 1.178, *p* = 0.301). Additionally, there was a significant age–sex interaction in the proportion of the stomach to body weight (*F*_3,11_ = 7.735, *p* = 0.005), whereas no age–sex interaction was observed for the other characteristics.

The weight of the rumen and reticulum tissue was the largest, accounting for 80.88 ± 0.45% of the stomach tissue weight, followed by the omasum (9.77 ± 0.34%) and abomasum (9.35 ± 0.42%). The body weight had a significant effect on the weight of the rumen and reticulum (*r* = 0.556, *F*_1,23_ = 10.307, *p* = 0.004), but no significant effect on the weight of the omasum, abomasum, or the proportions of each part of the stomach (*p* > 0.086 in all cases). With an increase in the body weight, the weight of the rumen and reticulum increased linearly, while the weights of the omasum and abomasum were negatively correlated with body weight, but none of the above correlation trends were considered significant (Figure 4).

Between seasons, the weights of the omasum (including contents), abomasum tissue, and abomasum (including contents) of blue sheep differed significantly, whereas there were no significant differences among the remaining characteristics (Table 3).

The size of the feed passage in the stomachs of blue sheep decreased according to the following order: omasum–abomasum passage > reticulum–omasum passage > cardiac > orifice ileocecal > pylorus. Additionally, there was no difference between males and females. Further, the size of the omasum–abomasum passage (*r* = 0.384, *F*_1,25_ = 4.330, *p* = 0.048) and ileo-cecal (*r* = 0.338, *F*_1,26_ = 4.618, *p* = 0.041) orifice were positively correlated with the body weight of the sheep (Figure 5).

### 3.3. Rumen Surface Enlargement Factor of Blue Sheep

The rumen characteristic parameters of 34 blue sheep (18 with known sex and age) were measured. Sex had no significant effect on the rumen characteristics (ANOVA, *p* = 0.244), and the mastoid density in the abdominal wall decreased significantly with age (*r* = −0.494, *p* = 0.037, *n* = 18). There were significant differences in the mastoid density, length, and width in different regions of the rumen (*p* < 0.001). In addition, the overall surface enlargement factor of the rumen mucosa was 2.85 ± 1.37 (Table 4).

### 3.4. Intestinal Length and Weight of Blue Sheep

The total length of the intestinal tract of blue sheep was 22.79 ± 0.26 m (*n* = 60), which was 26.1 ± 0.5 times the body length (*n* = 26). The small intestine accounted for about 73.06 ± 0.29% of the total length of the intestine (*n* = 59). There was no significant correlation between the total length of the intestine, the length of each intestinal component, and the body length (Pearson, *r* = 0.233, *p* = 0.251, *n* = 26), and there was also no significant difference in the total length of the intestinal tract and the length of each component between the two sexes (*F*_1,42_ = 0.427, *p* = 0.517). There was no significant difference in the total length of the intestinal tract and the length of each component between the two sexes between different seasons (*F*_1,58_ = 2.052, *p* = 0.157). The total intestinal weight (including contents) of blue sheep was 1.38 ± 0.06 kg (*n* = 61), accounting for 4.09 ± 0.34% of the sheep’s body weight (*n* = 26, Table 5). There were significant differences between the small intestine (*F*_1,60_ = 9.671, *p* = 0.003), small colon, (*F*_1,60_ = 12.434, *p* = 0.001), and rectum weights (*F*_1,60_ = 5.716, *p* = 0.020) in different seasons, and the weights of the small intestine, small colon, and rectum were larger in the winter than in the summer. There was no significant difference in the weights of various parts of the intestinal tract between male and female individuals (*p* > 0.164). There were significant differences in the weights of the small colon tissue (*F*_1,60_ = 4.541, *p* = 0.037) and rectal tissue (*F*_1,60_ = 4.709, *p* = 0.034) between different seasons.

### 3.5. Distribution of Feed Particles in the Digestive Tract of Blue Sheep in Different Seasons

The rumen and reticulum contained the largest feed particles (x > 2 mm), which reached 27.7 ± 3.03% of the total feed particles and gradually decreased in the omasum. The content of relatively large feed particles (2.0 mm ≥ x > 1.0 mm) was the highest in the omasum, reaching 6.17 ± 1.79%. However, the content of medium feed particles (0.425 mm < x ≤ 1.0 mm) was the lowest in the rumen and reticulum (2.89 ± 0.77%), being significantly lower in the other digestive organs. The contents of relatively small feed particles (0.425 mm ≥ x > 0.25 mm) were significantly lower in the rumen and reticulum than in the digestive tract after the abomasum. In addition, small feed particles (0.25 mm ≥ x > 0.125 mm) showed no significant differences in the content of each digestive organ. Thus, the contents of feed particles of different sizes changed between seasons (Figure 6).

The large feed particle content showed remarkable differences between the rumen-reticulum, small intestine, and cecum in different seasons. For relatively large feed particles, there were significant differences only in the rumen and omasum between different seasons. Medium feed particles also showed significant differences between the rumen-reticulum, small intestine, and cecum in different seasons. For relatively small feed particles, a lower content was present in the rumen-reticulum than in other digestive organs, but the difference was not significant. For small feed particles, there was a prominent difference only in the omasum, whose content (16.05 ± 2.52%) was significantly higher in the summer than in the winter (6.44 ± 1.45%, Table 6).

## 4. Discussion

Our results validate those of numerous other studies [4,14,52]. Through the dissection of adult blue sheep samples, we recorded the structural parameters of the sheep’s digestive tracts, such as the stomach and intestinal tract, and explored the distribution of feed particles with different sizes in the digestive tract. Previous studies have shown that blue sheep should be classified as a medium-sized ruminant in terms of their body weight (32.2 ± 1.4 kg) [53], and the proportion of the stomach to body weight (2.23 ± 0.93%). The findings of the current study are in agreement with those of other studies concerning ruminants, such as the sika deer (*Cervus nippon* at 2.2%, indicating a similar proportion of stomach to body weight), black-tailed deer (*Odocoiteus hemionus*, 1.2%), white-tailed deer (*O. virginianus*, 1.9%), fallow deer (*Dama dama*, 3.0%), red deer (*Cervus canadensis*, 3.0%), roe deer (*Capreolus capreolu*, 2.8%), and goat (*Capra hircus*, 2.7%) [52,54,55,56,57,58].

The weight of the organs that compose the gastrointestinal tracts of ruminants can vary based on the nutrition and type of feed consumed by those ruminants [59]. When the weight of the gastrointestinal tract increases, this is because the energetic demands of the tract also increase due to the weight-specific energy requirements of the tissue that remain unchanged [60]. Furthermore, a heavier gastrointestinal tract likely imposes a more substantial metabolic cost on the animal due to the high energetic demands of this tissue [61]. The implication is that animals should not commit to increases in the weight of their gastrointestinal tract organs unless there is a return from ingested nutrients or such a change is vital for their digestion and acclimation to local conditions [13]. The rumen and reticulum are the most important organs in ruminant stomachs and are mainly used to store and ferment feed. The force needed for rumen motility might require a thicker, more muscular tunic, which would lead to a heavier rumen-reticulum weight [62].

The weight of the rumen and reticulum tissue of blue sheep accounted for 80.88 ± 0.45% of the weight of the stomach tissue, which is similar to that of roe deer (70%), black-tailed deer (71%), red deer (77%), and sika deer (75%) [52,55,56,58]. The weight of the rumen and reticulum (contents included) accounted for 12.8 ± 0.04% of the body weight of blue sheep, ranging from 11% to 20%. The total length of the intestinal tract of blue sheep was 22.79 ± 0.26 m (*n* = 60); the small intestine accounted for about 73.06 ± 0.29% (*n* = 59) of the total intestinal length, and the weight of the small intestine accounted for 1.91% of the body weight. Based on the analysis of the length and weight of the gastrointestinal tract, blue sheep should be classified as rough feeders, which is in line with Hofmann’s theory [14]: most small ruminants adopt the energy strategy of a concentrate selector, foraging the necessary feed resources that have high nutritional quality and are easy to digest; on the contrary, large ruminants have high energy requirements and require a large amount of feed, which causes them to adopt the nutritional strategy of roughage feeders. The gastrointestinal tracts of roughage feeders are relatively long, which extends the retention time of feed particles to enable more efficient digestion and absorption of nutrients [47,52]. The case of the blue sheep in this study, with low rumen and reticulum contents in the summer, increasing rumen and reticulum contents towards the beginning of winter, with a subsequent decrease in the rumen and reticulum contents toward the end of the winter, might represent an example of such a shifting pattern in the forage quality and quantity. The weight of the intestinal tissue and content of blue sheep was higher in the winter than in the summer. The diet of blue sheep is primarily grass, which often includes *Poaceae*, *Ulmaceae*, and *Rosacea.* In the winter, blue sheep mainly feed on shrubs and trees [40], and the gut thickens to accommodate the high fiber content of this winter diet.

The wall of the rumen-reticulum is lined with papillae, a serous membrane, and a muscular tunic [17]. The papillae contributes to the nutrient absorption, the active transport of sodium and chloride, and the passive transport of volatile fatty acids and water [17]. The papillae of the ruminal mucosa vary among ruminants along with the season, diet, or feeding type [4,63,64]. Concentrate selectors should have higher SEF values than roughage feeders due to their selective intake of plant parts with higher nutrient contents [4,52]. Prins and Geelen [52] studied the rumen SEF values of fallow deer, red deer, and roe deer, and concluded that the SEF value of roughage feeders was less than 3.5, while that of concentrate selectors was between 7.0 and 9.5. Our study showed that the SEF of blue sheep was 2.85 ± 1.37, and the SEF of the vestibular area was significantly larger than that of other areas. The mastoid density in the rumen was significantly different in different regions, with the greatest density in the dorsal and ventral regions and a relatively lower density in the dorso-cecal and vestibular regions. This result is similar to those for the inhomogeneous distribution density of rumen papillae of the roughage feeders explained by Hofmann’s [4,52] theory, which further supports the classification of blue sheep as roughage feeders.

The digestibility of feed is determined by the excretion rate of feed particles of different sizes and the flow rate in the gastrointestinal tract, while the rate of excretion is determined by the feed type, particle size, and nutritional content [65]. A common feature of roughage feeders is that their feed is characterized by a high cellulose content and low nutritional quality, but they have a relatively large rumen and reticulum, which can better delay the passage of feed. Therefore, the feed particles in the gastrointestinal tracts of roughage feeders are relatively small. The proportion of feed particles smaller than 1 mm in the omasum of cattle and sheep is about 85~96%, with only 20~30% coverage in the rumen [66,67]. However, the feed strategy of concentrate selectors is different from that of roughage feeders, and the feed particles in the digestive tract of roughage feeders are often larger than those of concentrate selectors. For example, the proportion of feed particles larger than 1 mm in the omasum of moose (*Alces alces*) exceeds 50~60% [68]. Feed particles with a size larger than 1 mm in the omasum of blue sheep account for about 50% of the feed, which demonstrates that their nutrition feeding strategy is similar to that of moose. Therefore, the digestion mode of blue sheep is similar to that of concentrate selectors. The feed of roughage feeders is also high in fiber. In order to ensure that the feed is fully digested and absorbed by the rumen and reticulum, the reticulum-valve opening is relatively small, and only feed particles smaller than a certain standard size can pass through [47]. The pore size of the blue sheep feed passage decreases according to the following order: omasum–abomasum passage > reticulum–omasum passage > cardiac > orifice ileocecal > pylorus, with no significant differences between different sexes or seasons. Even though there are differences in the types and nutritional quality of feed consumed by blue sheep in different seasons, the size of each pore in blue sheep of different sexes remains relatively unchanged throughout the year. Therefore, it is difficult for blue sheep to delay the flow rate of large-sized feed particles in the rumen by adjusting the pore size of the rumen and reticulum.

The diameter of the feed passages in ruminants’ stomachs is the main factor that affects the feed flow rate [69]. In particular, the size and structure of the reticulum–omasum passage and omasum–abomasum passage play the most important role in regulating the flow rate of feed particles [70]. The reticulum–omasum passages of concentrate selectors are relatively large, and the feed particles that pass through have a relatively large size. However, the passages of roughage feeders are relatively small, and the feed particles that pass through them are, consequently, also relatively small. The reticulum–omasum passage of blue sheep (3.65 ± 0.11 mm) is slightly larger than that of Przewalski’s gazelle (3.05 mm), and cattle and sheep (2.91~3.25 mm), but smaller than that of moose (4.60~6.60 mm) [68,71]. The reticulum–omasum passage cannot adequately delay or block the passage of large feed particles through the omasum, so many large feed particles are still present after the omasum in blue sheep. The proportion of large feed particles in the rumen, reticulum, and omasum of blue sheep is higher in the winter than in the summer, but the opposite is true for small particles. In addition, the nutrient composition of the feed taken in by blue sheep varies greatly between different seasons, as the nutritional quality of the feed is higher in the summer than in the winter. Therefore, according to the theory of Kay and Hofmann, blue sheep prolong the flow time and fermentation time of feed particles in the stomach to fully digest and absorb feeds with a high fiber content. The nutritional quality of plants is closely related to phenological changes, and the nutritional quality of plants fluctuates greatly between seasons in the arid and semi-arid regions of Northern China. The nutrient quality of feed is the highest in the spring and summer and lowest in the winter. Blue sheep adopt different nutritional strategies to adapt to the environment and changes in the feed conditions.

## 5. Conclusions

The results obtained in the present study show that the digestive system of blue sheep in Helan Mountain, China, have the characteristics of typical ruminants, and they can be classified as medium ruminants in terms of their body weight. According to the morphological characteristics of the alimentary tracts of ruminants, the proportion of rumen and reticulum in the gastrointestinal tract, surface enlargement factors, and intestinal length and intestinal weight of blue sheep are consistent with the nutritional adaptations of roughage feeders. In this study, the pore sizes in the alimentary canal of blue sheep remained unchanged. However, the different grain sizes of feed particles in the digestive tract showed seasonal changes, indicating that the blue sheep’s digestive system is highly adaptable to environmental changes.

## Figures and Tables

**Figure 1 animals-13-01084-f001:**
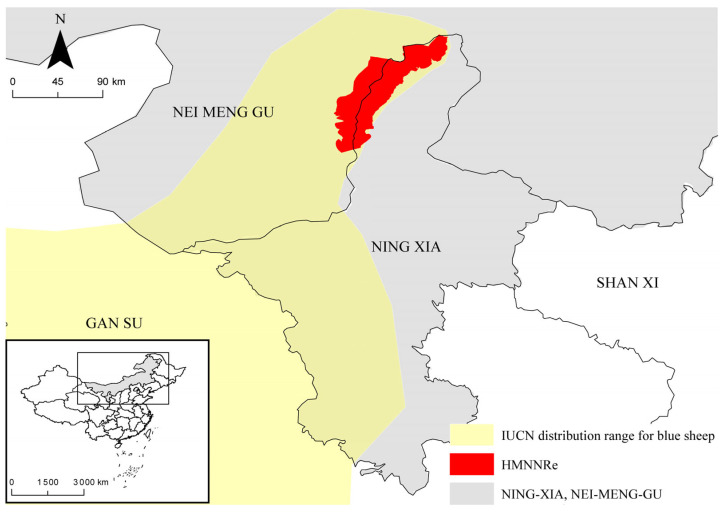
Map showing distribution range of blue sheep in HMNNRe including study area.

**Figure 2 animals-13-01084-f002:**
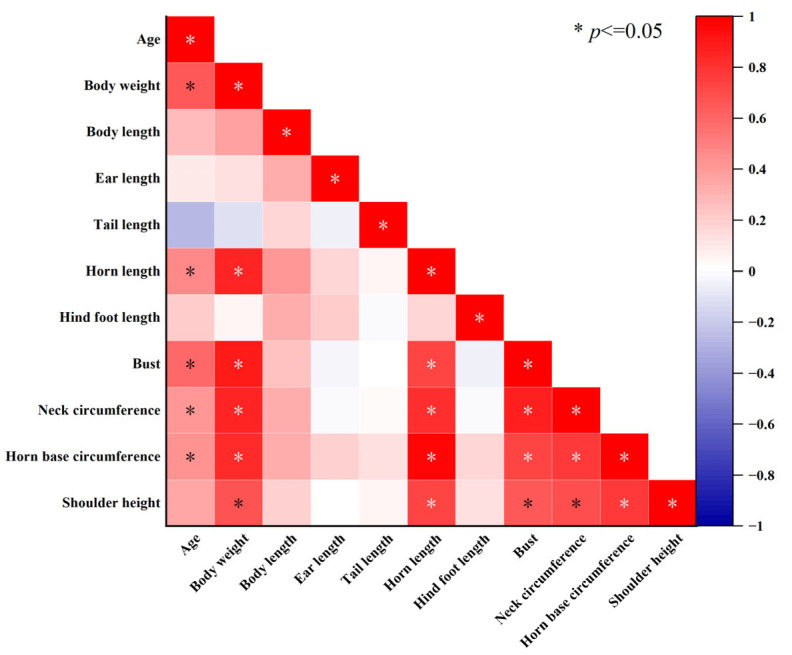
External morphological correlation of blue sheep.

**Figure 3 animals-13-01084-f003:**
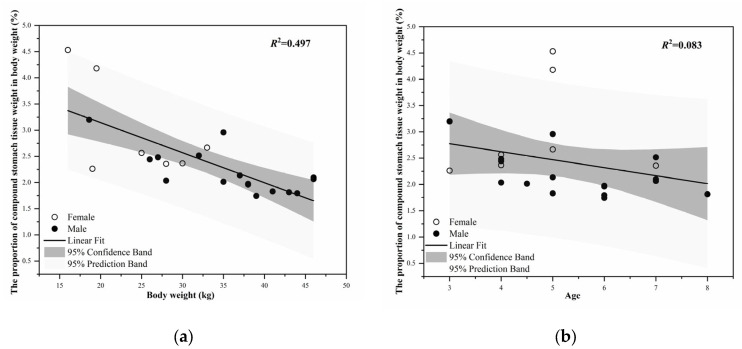
(**a**) The relationship between the proportion of stomach tissue weight to body weight and body weight; (**b**) The relationship between the proportion of stomach tissue weight to body weight and age.

**Figure 4 animals-13-01084-f004:**
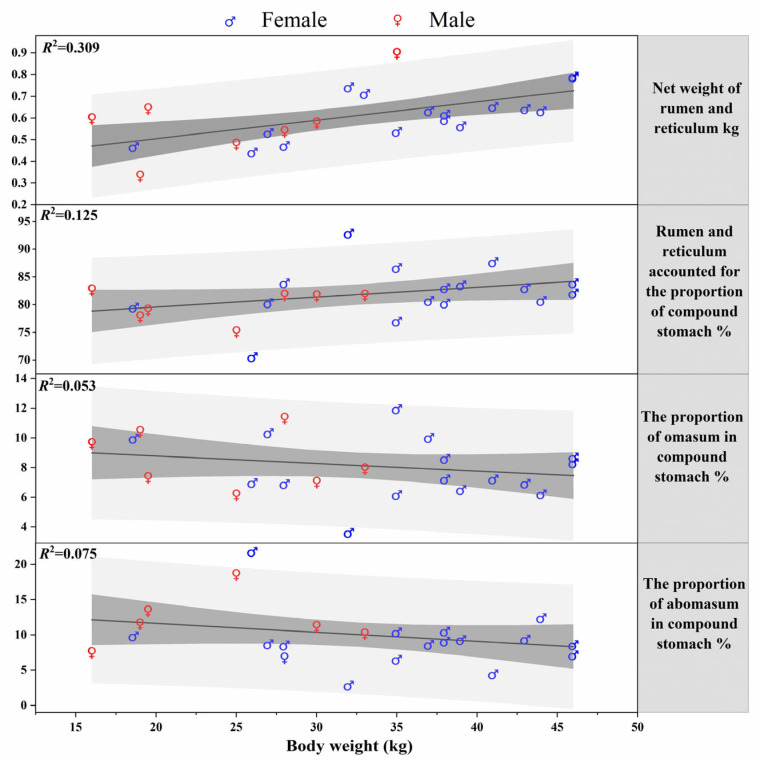
Relationship between each part of stomach and body weight.

**Figure 5 animals-13-01084-f005:**
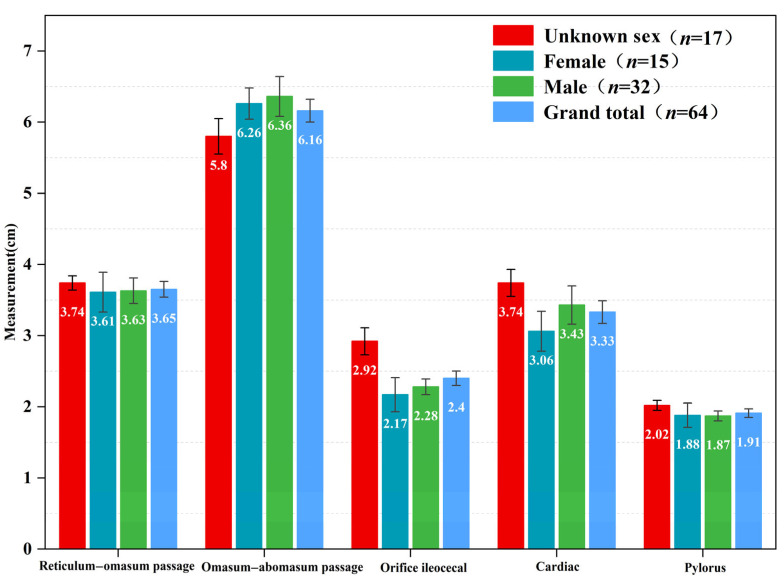
Measurements of various parts in compound stomach of blue sheep (cm).

**Figure 6 animals-13-01084-f006:**
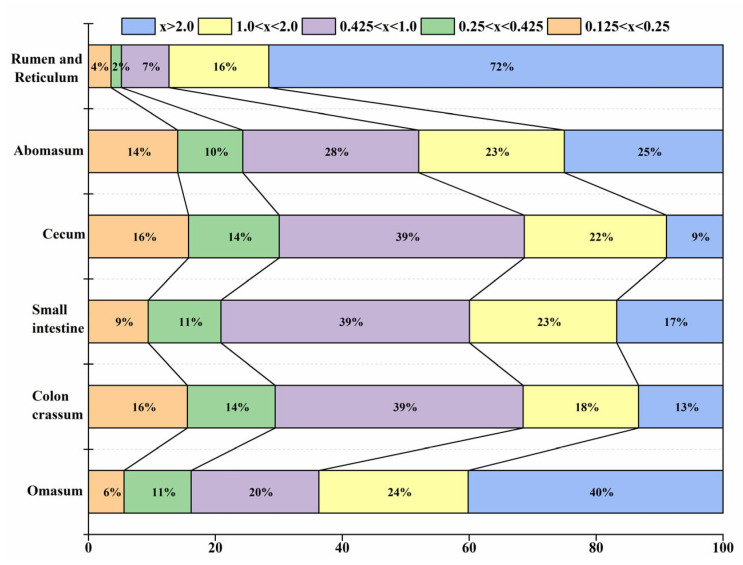
Distribution of different feed particles in different parts of the digestive tract of blue sheep.

**Table 1 animals-13-01084-t001:** External morphological characteristics and sexual two-state analysis of blue sheep.

Measurement Item	Male (*n* = 30)	Female (*n* = 11)	Grand Total (*n* = 41)	ANOVA/ANCOVA
Body weight (kg)	35.3 ± 1.4	23.8 ± 2.0	32.2 ± 1.4	*F*_1,38_ = 16.745, *p* < 0.001
Body length (cm)	91.2 ± 1.4	87.6 ± 1.6	90.2 ± 1.1	*F*_1,39_ = 1.953, *p* = 0.170
Ear length (cm)	10.7 ± 0.1	10.6 ± 0.4	10.7 ± 0.2	*F*_1,39_ = 0.039, *p* = 0.845
Tail length (cm)	10.2 ± 0.3	9.9 ± 0.3	10.2 ± 0.2	*F*_1,39_ = 0.365, *p* = 0.549
Horn length (cm)	41.5 ± 2.1	7.2 ± 1.0	32.3 ± 2.8	*F*_1,38_ = 88.862, *p* < 0.001
Hind foot length (cm)	27.4 ± 0.8	26.1 ± 0.7	27.1 ± 0.6	*F*_1,38_ = 0.048, *p* = 0.828
Bust (cm)	83.5 ± 1.4	73.0 ± 2.3	80.7 ± 1.4	*F*_1,38_ = 11.629, *p* = 0.002
Neck circumference (cm)	37.7 ± 1.4	26.6 ± 1.0	34.8 ± 1.3	*F*_1,38_ = 18.777, *p* < 0.001
Horn base circumference (cm)	22.8 ± 0.9	7.5 ± 0.8	18.7 ± 1.3	*F*_1,38_ = 97.624, *p* < 0.001
Shoulder height (cm)	70.5 ± 1.0	64.3 ± 1.0	68.8 ± 0.9	*F*_1,39_ = 12.278, *p* = 0.001

**Table 2 animals-13-01084-t002:** Weight of stomach tissue and the stomach including contents of blue sheep (kg).

Measurement Item		Unknown Sex	Male	Female	Total
Body weight		32.0 (1)	36.5 ± 7.4 (20)	24.3 ± 6.4 (7)	33.3 ± 8.7 (28)
Stomach	Tissue weight	0.72 ± 0.17 (17)	0.74 ± 0.28 (31)	0.79 ± 0.22 (15)	0.78 ± 0.03 (61)
	Including contents	1.92 ± 0.39 (17)	4.20 ± 0.39 (31)	4.31 ± 0.59 (15)	3.77 ± 0.28 (63)
Rumen and reticulum	Tissue weight	0.58 ± 0.14 (17)	0.66 ± 0.18 (29)	0.64 ± 0.19 (15)	0.63 ± 0.17 (61)
	Including contents	1.53 ± 0.38 (17)	3.79 ± 0.39 (31)	3.87 ± 0.58 (15)	3.19 ± 0.28 (63)
Omasum	Tissue weight	0.07 ± 0.04 (17)	0.07 ± 0.03 (32)	0.08 ± 0.03 (15)	0.07 ± 0.03 (64)
	Including contents	0.24 ± 0.01 (17)	0.26 ± 0.02 (31)	0.26 ± 0.03 (15)	0.26 ± 0.01 (63)
Abomasum	Tissue weight	0.06 ± 0.02 (17)	0.07 ± 0.02 (32)	0.07 ± 0.02 (15)	0.07 ± 0.02 (64)
	Including contents	0.14 ± 0.01 (17)	0.16 ± 0.01 (31)	0.18 ± 0.03 (15)	0.16 ± 0.01 (63)

Data are expressed as: mean ± standard error (sample size).

**Table 3 animals-13-01084-t003:** Weight of stomach and their weight including contents in different seasons (kg).

	Stomach		Rumen and Reticulum		Omasum		Abomasum	
	Weight of Tissue	Weight Including Contents	Weight of Tissue	Weight Including Contents	Weight of Tissue	Weight Including Contents	Weight of Tissue	Weight Including Contents
Spring	0.77 ± 0.04 (4)	2.61 ± 1.18 (4)	0.68 ± 0.08 (4)	2.40 ± 1.05 (4)	0.05 ± 0.02 (4)	0.13 ± 0.08 (4)	0.04 ± 0.02 (4)	0.08 ± 0.04 (4)
Summer	0.51 ± 0.08 (4)	3.68 ± 1.22 (4)	0.41 ± 0.07 (4)	1.78 ± 0.51 (4)	0.05 ± 0.01 (4)	0.09 ± 0.01 (4)	0.06 ± 0.01 (4)	0.08 ± 0.01 (4)
Autumn	0.73 ± 0.09 (4)	5.90 ± 2.51 (4)	0.56 ± 0.08 (4)	5.49 ± 2.32 (4)	0.05 ± 0.01 (4)	0.08 ± 0.01 (4)	0.11 ± 0.01 (4)	0.33 ± 0.20 (4)
Winter	0.78 ± 0.03 (17)	4.98 ± 0.37 (17)	0.63 ± 0.03 (17)	4.58 ± 0.32 (17)	0.06 ± 0.01 (17)	0.24 ± 0.01 (17)	0.07 ± 0.01 (17)	0.15 ± 0.01 (17)
ANOVA (*F*)	*F*_4,29_ = 2.66, *p* = 0.08	*F*_4,29_ = 1.72, *p* = 0.20	*F*_4,29_ = 2.57, *p* = 0.08	*F*_4,29_ = 3.06 *p* = 0.07	*F*_4,29_ = 0.91, *p* = 0.46	*F*_4,29_ = 9.57, *p* < 0.001	*F*_4,29_ = 4.30, *p* = 0.02	*F*_4,29_ = 4.63, *p* = 0.01

Data are expressed as: mean ± standard error (sample size).

**Table 4 animals-13-01084-t004:** The mastoid width, height, number, and surface enlargement factor of blue sheep.

Organ	Papillae Number (/2 cm^2^)	Papillae Height (cm)	Papillae Width (cm)	Surface Enlargement Factor
Atrium ruminis	55 ± 19 ^a^	0.263 ± 0.093 ^b^	0.105 ± 0.035 ^b^	3.90 ± 1.58 ^b^
Dorsal rumen	78 ± 26 ^b^	0.162 ± 0.070 ^a^	0.075 ± 0.028 ^a^	2.35 ± 0.96 ^a^
Ventral rumen	76 ± 25 ^b^	0.166 ± 0.063 ^a^	0.085 ± 0.027 ^a^	2.53 ± 0.94 ^a^
Dorsal rumen blindsacs	66 ± 25 ^ab^	0.179 ± 0.073 ^a^	0.081 ± 0.037 ^a^	2.62 ± 1.33 ^a^

The same letter in the superscript of the same column means that the difference is not significant (*p* > 0.05).

**Table 5 animals-13-01084-t005:** Intestinal length and weight between sexes in blue sheep in different seasons.

		Length					Weight				
		Small Intestine	Cecum	Colon Crassum	Microcolon	Rectum	Small Intestine	Cecum	Colon Crassum	Microcolon	Rectum
Sex	Unknown sex	1671.4 ± 42.4	34.5 ± 1.9	79.2 ± 6.6	366.5 ± 19.7	129.4 ± 8.6	1714.9 ± 22.8	1684.7 ± 42.5	1669.9 ± 19.9	1714.9 ± 22.8	1684.7 ± 42.5
	Male	1714.9 ± 22.8	33.3 ± 1.2	61.2 ± 2.1	397.6 ± 12.6	137.1 ± 4.8	33.3 ± 1.2	33.2 ± 1.2	33.6± 0.7	33.3 ± 1.2	33.2 ± 1.2
	Female	1684.7 ± 42.5	33.2 ± 1.2	70.9 ± 9.2	380.1 ± 14.1	124.1 ± 8.7	61.2 ± 2.1	70.9 ± 9.2	67.9 ± 3.0	61.2 ± 2.1	70.9 ± 9.2
	Total	1669.9 ± 19.9	33.6 ± 0.7	67.9 ± 3.0	384.9 ± 8.7	132.1 ± 3.8	397.6 ± 12.6	380.1 ± 14.1	384.9 ± 8.7	397.6 ± 12.6	380.1 ± 14.1
Season	Winter	1671.4 ± 21.1	33.9 ± 0.7	68.0 ± 3.2	386.4 ± 9.7	135.1 ± 3.6	0.51 ± 0.02	0.18 ± 0.02	0.21 ± 0.02	0.28 ± 0.01	0.24 ± 0.01
	Summer	1642.4 ± 56.3	28.5 ± 2.8	66.0 ± 8.4	368.3 ± 32.7	97.2 ± 17.3	0.31 ± 0.06	0.12 ± 0.04	0.16 ± 0.04	0.15 ± 0.02	0.14 ± 0.03
	ANOVA	*F*_1,61_ = 0.153, *p* = 0.697	*F*_1,61_ = 0.153, *p* = 0.697	*F*_1,61_ = 0.153, *p* = 0.697	*F*_1,61_ = 0.153, *p* = 0.697	*F*_1,61_ = 0.153, *p* = 0.697	*F*_1,61_ = 0.153, *p* = 0.697	*F*_1,61_ = 0.153, *p* = 0.697	*F*_1,61_ = 0.153, *p* = 0.697	*F*_1,61_ = 0.153, *p* = 0.697	*F*_1,61_ = 0.153, *p* = 0.697

The weight of each part of the intestine is the weight of the content.

**Table 6 animals-13-01084-t006:** Distribution and difference of feed particles in digestive tract of blue sheep in different seasons.

Organ	Season	Particle Size Distribution of Different Grades (%)				
		0.125 < x < 0.25	0.25 < x < 0.425	0.425 < x < 1.0	1.0 < x < 2.0	x > 2.0
Rumen and Reticulum	Summer	5.75 ± 1.00	3.42 ± 0.80	8.93 ± 0.83	4.20 ± 1.31	77.69 ± 2.02
	Winter	4.17 ± 1.07	2.71 ± 0.96	9.80 ± 1.80	17.97 ± 6.47	65.35 ± 7.17
	ANOVA	*F* = 0.33, *p* = 0.57	*F* = 0.08, *p* = 0.78	*F* = 0.04, *p* = 0.85	*F* = 6.69, *p* = 0.02	*F* = 4.44, *p* = 0.04
Omasum	Summer	16.05 ± 2.52	12.98 ± 3.89	25.46 ± 6.14	16.71 ± 7.32	28.80 ± 16.82
	Winter	6.44 ± 1.45	10.06 ± 2.44	26.99 ± 4.42	25.17 ± 3.78	31.34 ± 6.35
	ANOVA	*F* = 6.34, *p* = 0.02	*F* = 0.21, *p* = 0.65	*F* = 0.02, *p* = 0.90	*F* = 7.02, *p* = 0.01	*F* = 0.02, *p* = 0.89
Abomasum	Summer	12.55 ± 6.60	14.94 ± 4.68	18.76 ± 1.27	26.71 ± 5.64	27.04 ± 6.68
	Winter	16.76 ± 2.65	11.60 ± 1.96	29.57 ± 3.43	19.76 ± 3.75	22.32 ± 3.89
	ANOVA	*F* = 0.34, *p* = 0.56	*F* = 0.40, *p* = 0.53	*F* = 7.51, *p* = 0.02	*F* = 0.05, *p* = 0.49	*F* = 0.21, *p* = 0.65
Colon crassum	Summer	17.33 ± 1.34	14.88 ± 5.37	31.66 ± 8.17	13.25 ± 5.68	22.88 ± 8.12
	Winter	15.85 ± 2.49	12.50 ± 1.60	37.28 ± 3.40	18.29 ± 3.21	16.09 ± 3.77
	ANOVA	*F* = 0.05, *p* = 0.82	*F* = 0.28, *p* = 0.60	*F* = 0.38, *p* = 0.55	*F* = 0.35, *p* = 0.56	*F* = 0.46, *p* = 0.51
Small intestine	Summer	18.56 ± 4.39	9.71 ± 2.26	19.89 ± 6.71	22.07 ± 4.02	29.77 ± 8.24
	Winter	11.63 ± 2.52	11.25 ± 1.41	38.74 ± 3.23	24.49 ± 2.07	13.89 ± 3.03
	ANOVA	*F* = 1.09, *p* = 0.31	*F* = 0.17, *p* = 0.68	*F* = 4.80, *p* = 0.04	*F* = 0.20, *p* = 0.66	*F* = 3.95, *p* = 0.04
Cecum	Summer	16.06 ± 2.85	12.96 ± 2.75	26.22 ± 5.35	23.15 ± 4.89	21.61 ± 6.28
	Winter	18.12 ± 2.21	14.14 ± 1.53	35.93 ± 2.54	20.85 ± 3.12	9.96 ± 1.98
	ANOVA	*F* = 0.13, *p* = 0.72	*F* = 0.29, *p* = 0.60	*F* = 9.05, *p* = 0.01	*F* = 0.08, *p* = 0.78	*F* = 4.47, *p* = 0.03

## Data Availability

See the data of this study: https://datadryad.org/stash/share/1wG-CYDhthMlH-k9IKrVHgj7qU5CenbNH9mfgZIiNLc, accessed on 24 November 2022.

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
