# Peer review of "The Effect of Sexes and Seasons on the Morphological Structures of the Ruminant Digestive System of Blue Sheep (Pseudois nayaur)"

_animals, 2023, doi:10.3390/ani13061084_

Round 1

Reviewer 1 Report

No doubt there are some valuable pieces of information in the dataset acquired; these could be published, but will require the authors to identify a specific novel question they can answer with these data, and perform a limited set of analysis to test the emergent hypotheses and construct a focused narrative about. One possible question could be whether, how and why the ruminant SEF fluctuates seasonally, but for novel insights a diet analysis will also be required.

Reviewer 2 Report

General comments to authors

1-      Lines 35-36 should be removed from the abstract

2-      Line 40  they =  authors should mention  blue sheep

3-      Keywords should be reordered according to alphabetical order Please

4-      Line 41 food = should be feed

5-      Line 49 food = should be feed

6-      Line 52   reticulum, rumen, my question is does reticulum before rumen? I mean in the order? please clarify it

7-      Line 55 food = should be feed and please it will be at all the manuscript

8-      Line 62 food =should be feed and please it will be at all the manuscript

9-      Lines 79-81 needs references, please clarify

10-  Line 95 Blue sheep is a medium size wild goat? Does it sheep or goats? Please clarify

11-  Lines 95-96 these numbers should mentioned in which period and physiological statues?

12-  Line 112 the hypothesis of the study should be clear

13-  Line 113 aims to

14-  Line 118 me be? Please correct

15-  Lines 118-120 who told you the result promising, Hence the authors did not do the experiment yet? This sentence should be removed or modified to be mentioned at the hypothesize

16-  lines 152-154 sentence should be rewritten again

17-  Line 163  by the irrigation method, please clarify

18-  Line 163-167 some measurements needs references and more clarification for the reader

19-  Line 181 was= were. And please correct all these word food to be feed

20-  Line 182 the mixer of? Of what? Please clarify

21-  Samples collected in which seasons? Please clarify

22-  Line 199-201 the sentence not clear? should be removed

23-  Line 202 which control?

24-  Please the statistical analysis needs to be more simple and should mention the p-value and which level you used to compare the significant, furthermore which group of animals you considered as control ? Dose the authors considered the season as a variable factors at the analysis?

25-  Line 214 why authors did not mention that in the statistical analysis section (One-way variance (ANOVA) and covariance (ANCOVA) analysis)

26-  Line 227 significantly, which level?

27-  In my point of view, The results section is so long and  needs to rewrite again avoiding repeating all the numbers again once it's already located in the tables or even shown in the figures

28-  No data regarding the Environmental adaptations were presented why?

29-  At table 2 these numbers refer to what (1), (17), (31), (15), and (61) please clarify

30-  In my opinion, the conclusion that the authors mentioned as blue sheep

digestive system is highly adaptable to environmental changes. So what is the benefit I got? 

Author Response

Thank you very much for reviewing our manuscript and for your valuable comments and suggestions, which made us aware of the problems of our paper. Based on your comments and suggestions, we have revised the manuscript carefully.

Reviewer 3 Report

Dear Authors,

Thank you for submitting this manuscript that explores the digestive system morphology of the blue sheep. This is an interesting study and it has some potential use in the wider literature. However, the role of this study could be made much clearer. There is some scope for publication.

At current however, there seem to be some large revisions required in the manuscript to ensure the work is scientifically robust. I have attached the PDF version of the manuscript with specific comments. Additionally, please consider the following points: 

1. Methods. Please explain clearly how and why these samples were taken. This is a long time ago - why is this? What is the significance of summer versus winter? Were the animals used for other studies or purposes? The study methods need to be repeatable if this is to be puhblished.

2. Statistics. Pearson correlation and ANOVA are both forms of parametric test and assume normal distribution of data. Did you test your data for normality and if so what was the finding? Please report it in the work. If the data are not parametric an alternative test (such as Kruskal Wallis) would be more appropriate. If your data are not normal, you have violated the assumptions of the tests and your p values are inappropriate.

3. Wider application. Please explain further why these study findings matter.

4. Proof read. The work is challenging to read. A full proof read by a professional english speaker is necessary.

With these revisions, the work should be in a better position for consideration.

Round 2

Reviewer 2 Report

just I have one more question, the manuscript needs native language editing 

With my best regards 

Author Response

Dear Reviewer

Thank you very much for your comment. We used the language editing service provided by MDPI.

Reviewer 3 Report

Dear Authors,

Thank you for clearly addressing the comments arising from review.

Author Response

尊敬的审稿人

非常感谢您的宝贵建议,这对我非常有帮助。

亲切问候